# Effects of Different Phenolic Compounds on the Redox State of Myoglobin and Prevention of Discoloration, Lipid and Protein Oxidation of Refrigerated Longtail Tuna (*Thunnus tonggol*) Slices

**DOI:** 10.3390/foods13081238

**Published:** 2024-04-18

**Authors:** Suguna Palanisamy, Avtar Singh, Bin Zhang, Qiancheng Zhao, Soottawat Benjakul

**Affiliations:** 1International Center of Excellence in Seafood Science and Innovation, Faculty of Agro-Industry, Prince of Songkla University, Hat Yai 90110, Songkhla, Thailand; sugunap11699@gmail.com (S.P.); avtar.s@psu.ac.th (A.S.); 2Zhejiang Provincial Key Laboratory of Health Risk Factors for Seafood, College of Food and Pharmacy, Zhejiang Ocean University, Zhoushan 316022, China; zhangbin@zjou.edu.cn; 3College of Food Science and Engineering, Dalian Ocean University, Dalian 116023, China; qczhao@dlou.edu.cn; 4Department of Food and Nutrition, Kyung Hee University, Seoul 02447, Republic of Korea

**Keywords:** longtail tuna, phenolic compounds, metmyoglobin, oxymyoglobin, color, redness index value

## Abstract

Effects of different phenolic compounds on the redox state of myoglobin and their potential for preserving the color and chemical quality of refrigerated longtail tuna (*Thunnus tonggol*) slices were studied. Purified myoglobin from dark muscle (15.83 kDa) was prepared. Catechin, EGCG, quercetin, and hyperoside affected the absorption spectra and redox state of metmyoglobin (metMb) at 4 °C for up to 72 h differently. Reduction of metMb to oxymyoglobin (oxyMb) was notably observed for two flavonols (EGCG and quercetin) at 50 and 100 ppm. Based on the reducing ability of metMb, EGCG and quercetin were selected for further study. Longtail tuna slices were treated with EGCG and quercetin at 200 and 400 mg/kg. Color (a* and a*/b*), proportion of myoglobin content, and quality changes were monitored over 72 h at 4 °C. Tuna slices treated with 200 mg/kg EGCG showed better maintenance of oxyMb and color as well as lower lipid oxidation (PV and TBARS) and protein oxidation (carbonyl content) than the remaining samples. Nevertheless, EGCG at 400 mg/kg exhibited lower efficacy in retaining the quality of tuna slices. Thus, EGCG at 200 mg/kg could be used to maintain the color and prolong the shelf life of refrigerated longtail tuna slices.

## 1. Introduction

Tuna is a high-value species which contributes around 9.7% of the export value (USD 150 billion) of all aquatic products. Thailand has evolved to be an important contributor to the global tuna trade, since large tuna processing industries are fueled by raw materials directly landed in Thai ports by fishing fleets [1,2]. Tuna can be used for the production of several products, especially canned tuna. In addition, sushi and sashimi, the traditional Japanese food products prepared from thin slices of raw tuna with the highest quality, have become popular worldwide. In the Asia/Pacific regional markets, fresh and frozen tuna (non-canned) after the COVID-19 pandemic have been notably augmenting in demand and are more robust, compared to canned tuna. International trade for sashimi-grade tuna has been noticed to have the price rise by 20–30% [2]. Moreover, sashimi is not just ‘raw fish’, but rather a term that denotes a set of standards for texture, color, freshness, and presentation [3]. Color is a crucial index used for assessing freshness and acceptance of fresh tuna for sushi and sashimi products. After being sliced, the color of the flesh associated with myoglobin (Mb) and its derivatives cannot be maintained under refrigerated conditions for an extended period. Mb contributes to the color of fish flesh, as governed by its redox states involving oxymyoglobin (oxyMb), deoxymyoglobin (deoxyMb), and metmyoglobin (metMb), as well as their concentration [4,5]. In general, oxyMb contributes to the bright-reddish color of tuna flesh, whereas metMb has a brown color, which is mainly developed during handling and storage via autooxidation [6]. Furthermore, free radicals produced from lipid oxidation also induce the oxidation of Mb [7].

Mb is a heme protein that varies in amount of different muscles, according to fiber type, muscle activity, and animal age [8]. Mb has a large coiled polypeptide globin moiety and a heme prosthetic group. Eight α-helical domains of globin encircle the heme moiety in a coiled shape [9]. The protoporphyrin IX moiety consists of six coordination sites with central iron. Out of the six sites, four sites are constituted by nitrogen atoms of a protoporphyrin IX ring, and the fifth site is linked with the imidazole group of histidine, a part of globin. The changes in composition at the sixth site of heme iron determines meat color [10,11]. According to Singh et al. [12], the oxidized Mb is related to the oxidation of lipids and proteins, which results in an undesirable color and diminished appeal. Since dark muscle has more lipids than light muscle, it is more vulnerable to lipid oxidation than light muscle [13]. Fish muscle could be discolored because of lipid oxidation products that intensify Mb oxidation. Thus, it is important to maintain the oxyMb form to remain a desirable bright-red color.

Synthetic antioxidants usage is heavily regulated due to concerns over their potential health hazards, such as carcinogenesis. Moreover, consumers tend to reject synthetic food additives in general, further emphasizing the need for strict regulation of their use [14]. Phenolic compounds are known as naturally occurring antioxidants since they consist of at least one aromatic ring, which is attached to one or more hydroxyl groups. This allows them to retard or terminate reactive oxygen species-induced oxidative damage [15,16]. It has long been recognized that phenolic groups are good hydrogen donors and can create hydrogen bonds with protein C=O groups [17]. Epigallocatechin gallate (EGCG) is a water-soluble polyphenol with excellent electron and hydrogen transfer and metal-chelating ability [18,19]. Quercetin and hyperoside (quercetin 3-D-galactoside) are flavonoids present in several plants, such as tea, grapes, apples, and onions. Hyperoside is a glycoside derivative of quercetin and has a structure identical to quercetin, except for an O-glycosidic link connecting the galactoside group to the main structure [20]. Phenolic compounds generally have pronounced antioxidant ability; however, it depends on their structure as well as the number and position of hydroxyl groups [21]. Therefore, those phenolic compounds could be used to lower oxidation of tuna slices, thus retarding undesirable discoloration. Most phenolic compounds possess reducing power and can donate electrons to the central iron of the porphyrin ring in a Mb molecule, thus facilitating the reduction of Fe^3+^ to Fe^2+^. They indirectly safeguard Mb from oxidative damage by scavenging reactive oxygen species (ROS), which could oxidize Mb [22]. Different phenolic compounds might have varying abilities in providing the reduced Mb.

Chitooligosaccharide (COS) from squid pen could mitigate discoloration in yellowfin tuna slices. COS treatments (200–400 ppm) effectively reduced metMb formation and color changes during 9 days of storage at 4 °C, regardless of modified atmosphere packaging (MAP) conditions [23]. Specific phenolic compounds may stabilize oxyMb by preventing the formation of metMb, which helps meat to maintain its desired bright-red color [9,24]. Recently, Chung et al. [25] used several reducing agents to stabilize oxygenated equine heart Mb. Among those, quercetin-treated metMb solutions increased the reduction of metMb to oxyMb and enhanced the stability of the oxyMb. In our study, Mb from dark muscle of longtail tuna was isolated and characterized. The impact of several phenolic compounds on the reduction of metMb during storage at 4 °C for 72 h was also studied. In addition, the influence of selected phenolic compounds on color, proportion of Mb content, and chemical quality of longtail tuna slices was investigated during 72 h of storage at 4 °C.

## 2. Materials and Methods

### 2.1. Chemicals

All chemicals were of analytical grade and purchased from Sigma-Aldrich, Inc. (St. Louis, MO, USA). Low molecular protein markers were procured from GE Healthcare (Chicago, IL, USA). Quercetin (3, 3′, 4′, 5, 7-pentahydroxyflavone) with a purity of >95% was acquired from Yuanye Biotechnology Co., Ltd. (Shanghai, China). Hyperoside (quercetin 3-O-galactoside) was acquired from Zelang Biological Technology Co., Ltd. (Nanjing, China). Catechin and EGCG were purchased from Chengdu Biopurify Phytochemicals Ltd. (Sichuan, China).

### 2.2. Collection of Tuna Sample

Longtail tuna (*Thunnus tonggol*) (2–2.5 kg/fish) caught from the Gulf of Thailand were transported to the fresh market in Hat Yai within 24–36 h after capture. Fish were brought in ice to the laboratory within 30 min. The fish were then washed, beheaded, eviscerated, filleted, and manually dissected. The dark meat was collected and minced before Mb extraction.

### 2.3. Purification and Characterization of Myoglobin from Longtail Tuna

For Mb extraction and purification, dark muscle mince (100 g) of longtail tuna was mixed with 300 mL of cold extracting medium (10 mM Tris-HCl, pH 8.0, containing 1 mM EDTA and 25 g/L Triton X-100), followed by homogenization for 120 s using an IKA Labortechnik homogenizer (Shah Alam, Selangor, Malaysia) [26]. Then, the homogenate was centrifuged at 5000× *g* for 10 min at 4 °C using an RC-5C plus centrifuge (Allegra 25-R, Beckman Instruments Inc., Palo Alto, CA, USA). The supernatant was vacuum filtered with two Whatman filter papers no. 1. Filtrate was adjusted to pH 8.0 using 0.2 M NaOH. Solid ammonium sulfate (65–100%) was added to the filtrate for precipitation. The obtained precipitate was subsequently dissolved in a minimal volume of cold starting buffer (5 mM Tris-HCl buffer, pH 8.0, containing 1 mM EDTA). Then, the mixture was dialyzed against 10 volumes of starting buffer for 24 h at 4 °C. After dialysis, the dialysate was further loaded on a Sephadex G-50 column (2.6 × 70 cm; Amersham Bioscience, Uppsala, Sweden), previously equilibrated with starting buffer. Proteins were separated at a flow rate of 0.5 mL/min and the fractions (3 mL) were collected and monitored at 280 and 540 nm. The fractions with the peak at 540 nm representing Mb were pooled and characterized.

#### 2.3.1. Sodium Dodecyl-Sulfate Polyacrylamide Gel Electrophoresis (SDS-PAGE)

Molecular weight (MW) and the purity of Mb were examined by SDS-PAGE [27]. Fifteen µg of protein sample was loaded on polyacrylamide gel (4% stacking gel; 17.5% running gel). After separation, the proteins were subjected to staining and destaining, respectively. MW of the target protein band was then computed via the plot of the retention factor and log (MW) of the protein standard. The band intensity was measured using a densitometer (modal GS-700) with the aid of Molecular Analyst Software version 1.4 (Bio-Red Laboratories, Hercules, CA, USA).

#### 2.3.2. Preparation of metMb

The conversion of Mb to metMb was carried out [5]. Potassium ferricyanide (5 mg/mL) was mixed with Mb solution and continuously stirred for 1 h in an iced bath. Residual ferricyanide was removed using a Vivaspin^®^ 20 centrifugal concentrator. The desalted metMb solution was diluted to 0.6 mg/mL with cold 50 mM phosphate buffer (pH 8.0).

#### 2.3.3. Absorption Spectra and Proportion of Mb Forms

Absorption spectra and the Soret band of metMb solution were determined using a UV-Vis spectrophotometer (UV-1601, Shimadzu, Kyoto, Japan) [28]. The measurement was run from 350 to 750 nm at a scanning rate of 1000 nm/min and 40 mM phosphate buffer; pH 6.8 was used as blank [29].

The proportions of oxyMb, metMb, and deoxyMb were computed by a modified Krzywicki’s equation [5]:[OxyMb] = (0.722R1 − 1.432R2 − 1.659R3 + 2.599) × 100
[MetMb] = (−0.159R1 − 0.085R2 + 1.232R3 − 0.520) × 100
[DeoxyMb] = (−0.543R1 + 1.594R2 + 0.552R3 − 1.329) × 100
where, R1 = A_582_/A_525_, R2 = A_557_/A_525,_ and R3 = A_503_/A_525_.

### 2.4. Effect of Phenolic Compounds on the Proportion of Mb Forms in metMb Solution

The solution of metMb (final concentration of 0.6 mg/mL) was treated with various phenolic compounds (catechin, EGCG, quercetin, and hyperoside) having different structures (Figure 1) at varying final concentrations (50 and 100 ppm). Phenolic compounds were first dissolved in 50 mM phosphate buffer (pH 7.0) before adding to the metMb solutions. The pH of Mb solutions was brought to 7.0 for all samples and kept at 4 °C for 72 h. Solutions (1.2 mL) were taken at 0, 36, and 72 h for analysis.

#### 2.4.1. Absorption Spectra and Proportion of Mb Forms

The absorption spectra and proportions of Mb forms in metMb solution treated with phenolic compounds were measured as mentioned in Section 2.3.3. The proportions of oxyMb, metMb, and deoxyMb were recorded.

#### 2.4.2. Tryptophan Fluorescence

The tryptophan fluorescence of the metMb solutions treated with different phenolic compounds was measured with the aid of a spectrofluorometer (RF-15,001, Shimadzu, Kyoto, Japan). An excitation wavelength of 280 nm and an emission wavelength of 325 nm were used [30].

Among four phenolic compounds, EGCG and quercetin were able to maintain the oxyMb in the metMb solution more effectively. Hence, they were selected for further study.

### 2.5. Effect of EGCG and Quercetin on Mb, Color, and Lipid/Protein Oxidation of Refrigerated Longtail Tuna Slices

#### 2.5.1. Preparation of Tuna Slices Treated with EGCG or Quercetin

The dorsal part was cut into slices (length: 6–8 cm, width: 3–4 cm, thickness: 0.5 cm). Twenty-five grams of slices were treated with EGCG or quercetin to obtain final concentrations of 200 or 400 mg/kg slices. Phenolic compounds were solubilized in the minimum volume of distilled water containing 0.02% sodium azide, which prevented microbial spoilage during the storage period (2 mL for 100 g slices). Solution was then applied on the surface of slices using a sterilized brush, followed by drying in a sterile cabinet (15 min, 25 °C) [31]. Subsequently, slices (25 g) were displayed on a polystyrene foam tray, which was further inserted into a multilayer low-density polyethylene bag (8 × 12 inch^2^) and sealed using a hand sealing machine. The slices treated with EGCG or quercetin at 200 and 400 mg/kg slices were denoted as E-200 and E-400 or Q-200 and Q-400, respectively. The slices without treatment were named as the control. During the storage at 4 °C, the samples were randomly taken at 0, 12, 24, 36, and 72 h for analysis. For fatty acid profile analysis, untreated and treated samples stored for 72 h were compared with a fresh sample (without any treatment).

##### Myoglobin Forms and Color

Firstly, a sample (1 g) was homogenized (10,000 rpm, 10 s) in 10 mL of cold phosphate buffer (40 mM, pH 6.8). The homogenate was centrifuged (8000× *g*, 30 min, 4 °C). The supernatant was subjected to filtration with Whatman filter paper no. 1. The filtrate was used for spectral analysis (350–750 nm) against the same buffer. The relative percentage of Mb derivatives was computed as detailed in Section 2.3.3.

Lightness (L*), redness (a*), and yellowness (b*) were measured using a colorimeter (HunterLab, Model ColorFlex, VA, USA). The slices were subjected to 4 readings from different parts of a slice. The redness index (a*/b*) of the meat was examined as tailored by Chen et al. [32].

##### Lipid Oxidation

Peroxide value (PV) measurement was conducted using the ferric thiocyanate method [33]. A standard curve of cumene hydroperoxide (0.5–2 ppm) was prepared. PV was computed from a standard curve and expressed as mg cumene hydroperoxide per kg meat after blank subtraction.

The TBARS value was examined as detailed by Benjakul and Bauer [34]. The TBARS value was computed from the malonaldehyde (0–5 ppm) standard curve and reported as mg malonaldehyde (MDA)/kg of tuna meat.

##### Total Sulfhydryl Content

Total sulfhydryl content was determined using 5,5′-Dithiobis (2-nitrobenzoic acid) (DTNB) [35]. The content was computed using a molar extinction coefficient of 13,600 M^−1^cm^−1^ and the concentration was reported as µmol/mg protein.

##### Protein Carbonyl Content

Protein carbonyl content was determined by derivatization with DNPH (2,4-dinitrophenyl hydrazine) according to Nikoo et al. [36]. The carbonyl concentration (µmol/mg protein) was calculated using an absorption coefficient at 370 nm of 22,000 M^−1^ cm^−1^.

##### Fatty Acid Composition

Firstly, oil was extracted by the Bligh and Dyer method [37]. Then, oil (10 mg) was subsequently subjected to the preparation of fatty acid methyl esters (FAME) via transmethylation [38]. FAMEs were separated and quantified using gas chromatography equipped with a flame ionization detector (Agilent 7890B, Santa Clara, CA, USA). Fatty acids were determined and reported as mg/g oil.

### 2.6. Statistical Analysis

Three distinct sample lots were utilized in this study. All experiments and analyses were run in triplicate. The data underwent a one-way analysis of variance (ANOVA), followed by a comparison using Duncan’s multiple range test. Statistical Package for Social Science (IBM SPSS statistics 25, Armonk, NY, USA) was employed.

## 3. Results and Discussion

### 3.1. Purification of Myoglobin from Longtail Tuna Dark Muscle

Based on A_280_, two major peaks were obtained after separation using the Sephadex G-50 column (Figure 2A). The second peak showed a brown color representing Mb as the dominant constituent. This was coincidental with the appearance of a peak with A_540_ representing Mb [39]. When crude Mb extract and all the fractions were analyzed for their protein patterns as shown in Figure 2B, the crude Mb extract possessed several protein bands with MWs of more than 30 kDa (Figure 2B, lane 2). In addition, the band with low intensity having a MW of 15.83 kDa was also observed in the crude fraction. For the ammonium sulphate (AS) fraction, proteins with MWs above 66 kDa were eliminated, whereas the proteins with MWs of 15.83 kDa became more intense (Figure 2B, lane 3). Further purification was achieved through the Sephadex G-50 column. Pooled fractions (fraction Nos. 20–25) appeared as a single band, indicating the homogeneity of the purified Mb. The MW of Mb from longtail tuna dark muscle was 15.83 kDa. The obtained MW was comparable to that of other tuna species such as Eastern little tuna (15.68 kDa), bigeye tuna (15.54 kDa), and yellowfin tuna (16 kDa), but is slightly larger than that of Snake-head fish (15 kDa), mackerel (14.90 kDa), and sardine (14.60 kDa) [39,40,41,42,43]. MWs of porcine, ovine, and sperm whale Mbs were 17.70 kDa, 17.10 kDa, and 17 kDa, respectively. Renerre et al. [44] documented that bovine Mb possessed a MW of 17 kDa. Fish Mb typically has a lower MW than mammalian Mb due to differences in metabolic demands and oxygen requirements between aquatic and terrestrial environments [45].

According to the densitometric analysis, the band intensity of Mb in the Sephadex G-50 fraction accounted for almost 91.03% of the total protein. Chaijan et al. [46] obtained 92% of purity through three steps of purification. Sephadex G-50 is suitable for the separation of molecules having MWs ranging from 1.5 to 30 kDa. On the other hand, Sephadex G-75 is appropriate for proteins possessing MWs in the range of 3–80 kDa. Thus, Sephadex G-50 was suitable for the purification of Mb with high purity.

### 3.2. Absorption Spectra and Proportion of metMb Solutions

The spectrum of metMb from longtail tuna dark muscle in the ranges of 350–750 nm is depicted in Figure 3. MetMb solution showed high peaks in the blue area (350–450 nm), representing the Soret band at 406 nm. Similar results were observed in metMb purified from yellowfin tuna [47]. The Soret band of metMb purified from longtail tuna was noticeable at 407 nm [30]. The Soret band, caused by the interaction between heme and apomyoglobin, indicates the unfolding of hemoproteins [48]. In the region of 450–750 nm, the peaks were attained at 502 and 630 nm for the metMb solution (Figure 2). According to Nurilmala et al. [49], metMb of tuna had maximum absorbance wavelengths at 502 and 631 nm. Moreover, Mb from equine heart and horse skeletal muscle exhibited a similar spectral pattern [5,50], with the peaks around 503 and 632 nm for metMb. It has been reported that the three redox forms of Mb displayed distinct absorption spectra, showing an isobestic point at 525 nm [5]. The isobestic point at 525 nm is selected as a reference point because it corresponds to a situation where the millimolar extinction coefficients of the three Mb forms are equivalent, facilitating an accurate determination of overall proportions [5,51,52]. The wavelengths of 503, 557, and 582 nm are corresponding to metMb, deoxyMb, and oxyMb, respectively [5]. In metMb solution, metMb was predominant (85.22%), while deoxyMb (8.8%) and oxyMb (5.97%) were present at lower proportions. Q bands provide a means to monitor changes in the ligation state of the heme iron [53]. The changes in the conformation or structure of the porphyrin ring as influenced by the electronic transitions are responsible for the alteration of Q bands in the absorption spectrum [54,55]. The presence of Q bands indicated the integrity of the porphyrin ring of purified Mb (Figure 3).

### 3.3. Effect of Phenolic Compounds on Absorption Spectra and Proportion of Different Forms in metMb Solution

#### 3.3.1. Absorption Spectra

The Soret bands of all samples were observed between the range of 405 and 406.5 nm, indicating the stability of porphyrin moiety connected to the globin, regardless of type and concentration of phenolic compounds (Figure 4). The Soret band originates from the π~π* transition of the porphyrin ring of heme and directly related to the heme content [50]. Similarly, the Soret band was found at 407 nm for metMb solutions purified from Eastern little tuna [30]. Different red shifts in the spectra of these samples could be due to varying structures as affected by different phenolic compounds. A blue shift from 406.5 to 404 nm after 72 h was in line with the upsurge in metMb content, especially in the control. The disappearance of the Soret absorption peak suggested that heme protein was destroyed, or the porphyrin moiety was separated from the globin [56]. Moreover, the absorption intensity of the Soret bands of the EGCG-treated metMb solution was increased with augmenting concentration. This result was opposite to the control and other treated samples as presented in Figure 4A–C. At the same time, the Q bands started disappearing and the disappearance was complete with increasing concentrations of EGCG as shown in Figure 4A–C.

The UV-visible spectra of the various samples generally exhibited different baselines. This discrepancy arose due to variations in the oxidation states of the Mb in different samples [29]. Inherent colors of phenolic compounds used could also contribute to such a difference to some extent.

#### 3.3.2. Proportion of Mb Forms

The effect of phenolic compounds at different concentrations on metMb from longtail tuna in terms of proportion of different forms is shown in Figure 5A. After the addition of phenolic compounds, catechin (50 and 100 ppm) increased the metMb content (*p* < 0.05). Similar metMb contents between the control and C-50 after 72 h of storage were observed (*p* > 0.05) but it was lower than C-100 (*p* < 0.05). When comparing differently treated samples, those treated with EGCG had lower metMb content, followed by those treated with quercetin (*p* < 0.05). As the time upsurged to 72 h, the metMb content became lowered for all the samples treated with phenolic compounds; however, C-100 sample had the maximum metMb content, whereas E-100 showed the lowest concentration of metMb (*p* < 0.05). This result indicated the highest reducing power of EGCG at higher concentrations. EGCG has good electron and hydrogen transfer and metal chelation properties [19]. According to Quan et al. [57], the gallate group at the third position of EGCG and the three OH groups at the B-ring function as free radical scavengers. Moreover, the EGCG had a higher reducing power than catechin [58]. However, metMb contents of H-50 and H-100 were similar to that of the control up to 36 h (*p* > 0.05) but were significantly reduced after 72 h compared to the control (*p* < 0.05). Quercetin was shown to effectively inhibit Mb oxidation in minced pork at a pH of less than 6.8 [59].

The impact of phenolic compounds at different concentrations on deoxyMb proportion in metMb solutions is displayed in Figure 5B. At the beginning (0 h), it was noted that all samples showed a deoxyMb concentration below 10% except EGCG-treated samples. Over the period, deoxyMb content of all the samples increased (*p* < 0.05). Notably, the E-100 sample reached the maximum value (above 20%), which was related to a lower proportion of metMb (Figure 5A,B). Overall, EGCG-treated samples had the maximum deoxyMb content. Quercetin-treated samples showed lower deoxyMb content than the EGCG-treated samples (*p* < 0.05). Conversely, catechin-treated samples showed a lower deoxyMb content than the control (*p* < 0.05) but possessed a higher deoxyMb content than those treated with hyperoside (*p* < 0.05).

At the beginning (0 h), the oxyMb content of metMb solution without and with treatment by various phenolic compounds was in the range of 2.92–13% (Figure 5C). Typically, oxygen reacts with Mb, forming a bright-red pigment known as oxyMb, which enhances visual appeal [7,60]. For the control, a drastic reduction in oxyMb content was attained after 72 h, when compared to 0 h (*p* < 0.05). It also had the least oxyMb content of all treated samples (*p* < 0.05). The phenomenon was most likely linked to the autoxidation process, in which oxyMb was converted to metMb. Quercetin-added metMb solution, especially at 100 ppm, had the maximum oxyMb content. Throughout the storage, samples treated with EGCG had lower oxyMb content than those added with quercetin (*p* < 0.05). Catechin-added samples showed lower oxyMb content than other treated samples (*p* < 0.05) and C-100 ppm had no significant difference with the control after 72 h (*p* > 0.05). The result coincided with the increased metMb content (Figure 5A,C). This was in line with the higher oxidation of oxyMb to metMb. After 72 h, phenolic compounds increased the oxyMb content (*p* < 0.05), depending on their concentration. Catechin-added samples had lower oxyMb content than samples treated with other phenolic compounds.

Overall, the oxidation of iron in the heme group of all Mb solutions upsurged. This was confirmed by the decrease in absorbance intensity of the spectrum of the control. Chung et al. [25] observed the changes of metMb from different species under certain conditions such as pH and temperature.

#### 3.3.3. Tryptophan Fluorescence Intensity

Tryptophan fluorescent intensity of metMb from longtail tuna treated without and with various phenolic compounds at different concentrations is shown in Figure 6. Phenolic compounds have the potential to interact with metMb, particularly via H-bond or hydrophobic interaction. The interaction with globin could affect its oxidation state and the stability of the porphyrin ring [7]. As a protein undergoes unfolding, amino acid residues previously shielded inside protein molecules become exposed to the aqueous solvent. Tyrosine and tryptophan residues are frequently among these exposed residues [30] and can be used for monitoring protein structural alteration [61]. Except for catechin-treated samples, all the treated samples had minimum changes in tryptophan fluorescence intensity, compared to the control. It was assumed that other phenolic compounds did not induce the unfolding of globin. The presence of a tryptophan residue indicated alterations in the tertiary structure of protein [62]. Tryptophan plays a key role in establishing favorable and specific tertiary interactions within the native structure of apomyoglobin [63]. The highest intensity in tryptophan fluorescence was noticeable for catechin-treated metMb samples (*p* < 0.05), especially at 100 ppm. Therefore, catechin could enhance the conformational changes of globin. At higher concentrations, catechin might interact with globin to a higher extent. Consequently, loss of protein structure and functionality might occur along with the metMb aggregation. This could also induce the oxidation of Fe^2+^ in the porphyrin ring as related to the increased metMb content in catechin-treated samples. Slight changes in the Soret band of catechin-treated samples were also found (Figure 4).

### 3.4. Effect of EGCG and Quercetin on Color, Mb, and Quality of Longtail Tuna Slices during Refrigerated Storage

#### 3.4.1. Color

The color of tuna slices treated without and with EGCG and quercetin at different concentrations during 72 h of storage was varied (Figure 7 and Figure 8). Snyder et al. [64] stated that the redness index (a*/b*) is a stronger predictor and much more appropriate than lightness (L*) and yellowness (b*) for bluefin or yellowfin tuna meat. Additionally, the redness index (a*/b*) is a more effective measure than redness (a*) for determining beef color [64]. Similarly, in the present study, a*/b* and a* values were better indicators than L* value for assessing the color of tuna slices. At the beginning (0 h), similar a*, a*/b*, and L* values were obtained for all the samples, which were found in the range of 7.48–7.63, 0.700–0.716, and 32.73–34.73, respectively (Figure 8A–C). Thus, the addition of both EGCG and quercetin had no notable impact on the color of tuna slices at the initial time (*p* > 0.05). The result was also substantiated by the appearance of tuna slices: as displayed in photographs, all treated samples and the control had the same bright-red color (Figure 7). Red meat in the anterior and central parts of skipjack tuna contained Mb content of 20 mg/g tissue [65]. In the case of Eastern little tuna, the dark and ordinary muscle contained Mb at 11.77 and 2.34 mg/g sample, respectively [30]. The Mb content in tuna muscle can vary, depending on factors such as species, age, size, muscle type, and physiological condition. In general, fish can be classified as light- and dark-fleshed fish based on the level of sarcoplasmic protein, especially due to the presence of Mb content. The dark-fleshed fish contains more lipids as compared to light muscle as well as meat from other mammals such as pork, beef, etc., and this makes it more prone to lipid oxidation relative to the light muscle [12,13]. Lipid oxidation products can further enhance the oxidation of Mb, thus causing discoloration of fish muscle. When compared with meat from mammals rich in Mb such as beef, the role of phenolic compounds in preventing lipid and Mb oxidation was presumed to be similar to tuna slices. Moreover, different fish species have variations in myoglobin content. It may have a difference in globin structure. These factors could determine the efficacy of the phenolic compounds in maintaining the color of varying fish meat in different ways. Additionally, chemical compositions in the muscle, especially lipid and polyunsaturated fatty acid content, associated with their lipid oxidation products, could affect the rate of Mb oxidation to different degrees. As a consequence, the efficiency of phenolic compounds in preventing discoloration of fish slices can be varied [14,31,57].

When comparing the appearance among the different treatments at 12 and 24 h, the E-200 sample had maximum redness (a* value) compared to the control, E-400, Q-200, and Q-400 (*p* < 0.05), in which Q-200 showed a lower a* value than other treatments (*p* < 0.05) but had a similar value to the control sample (*p* > 0.05) (Figure 7). This was plausibly due to the quinone formation from quercetin upon its oxidation. Quercetin has the capacity to donate electrons from the B ring and a half of the C ring [66]. In the presence of peroxidase, A ring present in quercetin oxidizes to o-quinone and further undergoes to form quinone methide isomers, which are more electrophilic and they themselves can act as prooxidants under certain circumstances [67]. However, catechol in quercetin can covalently bind to cysteine residues of proteins. This process involves converting the catechol to its quinone form, which then forms a covalent bond with the protein thiol, ultimately regenerating the bound catechol [68]. After 72 h of storage, the E-200 and Q-400 samples had higher redness (a*) than the control, E-400, and Q-200 (*p* < 0.05) (Figure 7 and Figure 8). Over the period, quercetin Q-400 had higher redness (*p* < 0.05) than Q-200 but lower than E-200 (*p* < 0.05). Furthermore, the highest redness was found in the E-200 sample (*p* < 0.05). When the concentration of EGCG increased, a* value was reduced (*p* < 0.05). Although EGCG is an effective reducing agent, its interaction with proteins plausibly caused rapid discoloration of the tuna slices, particularly at 400 mg/kg. Similar results were obtained in tuna slices treated with EGCG and COS, especially at higher concentrations used [31]. This was most likely related to a change in protein structure as induced by EGCG at 400 ppm. Thus, EGCG could expose the hydrophobic groups and interact with the proteins, particularly the sarcoplasmic proteins, leading to coagulation [69]. This might reduce the effectiveness of EGCG in converting metMb to oxyMb.

Overall, the interaction between phenolic compounds and globin or other muscle proteins could therefore determine the color alteration or stability of tuna slices, depending on type and concentration of phenolic compounds used. For the oxidation of Mb, phenolic compounds such as EGCG or catechin can bind with globin, thus stabilizing globin molecules. Also, with their reducing powder, they can reduce central ferric to ferrous in the porphyrin ring of Mb. However, the type and concentration of phenolic compounds used were the main factors affecting the oxidation or stability of Mb.

The redness index (a*/b*) of tuna slices treated without and with EGCG and quercetin at different levels for 72 h at 4 °C is shown in Figure 8B. The color of the slices, especially for sashimi products, determines acceptability. The ratio of a* and b* has been used as an indicator of change in redness [32] and to assess discoloration in tuna meat during storage [24]. The continuous decrease in redness index was found in E-400 and Q-200 samples as a function of time. Similar to redness, the highest redness index was noticed in the E-200 sample, followed by the Q-400-treated sample (*p* < 0.05). Over the period of storage, the quercetin Q-400 sample maintained a higher redness value (*p* < 0.05) than Q-200 but lower than E-200 (*p* < 0.05). At higher concentrations, quercetin might also participate in self-polymerization or complexation with metal ions, thus reducing the quinone production [70,71]. After 72 h of storage period, the samples treated with E-200 and Q-400 had maintained redness index (a*/b* value) more effectively than the control, E-400, and Q-200 (*p* < 0.05). Conversely, the Q-400 sample within the first 24 h of storage showed a lower redness index than E-200 (*p* < 0.05). Quercetin might have been oxidized rapidly. However, its oxidized form could have been reduced by indigenous reducing agents such as NADH or indigenous enzymes [72]. As a consequence, the redness became more pronounced at the end of the storage. Since quercetin was used at a higher level (400 mg/kg), it was not suggested for color maintenance. Furthermore, the decrease in the redness index was related to the darkening of meats, which resulted from the formation of metMb along with the disappearance of the Soret absorption band and the shift of the Soret peak (Figure 4 and Figure 5) [28,29]. This agreed well with the absorption spectra of phenolic-compound-treated metMb solutions during 72 h storage at 4 °C (Figure 4A).

After 72 h storage, the lightness of EGCG- and quercetin-treated samples was slightly increased from 0 h (Figure 8C). In contrast, the control had higher lightness. Notably, a*/b* value of the control decreased, and color fading was simultaneously observed rather than the darkening of slices (*p* < 0.05) (Figure 8A,B). Although the control had high metMb content, the augmented lightness value was associated with the fading of tuna slices (Figure 8C). This result was also substantiated by the appearance of the tuna slices in Figure 7. As Mb undergoes denaturation, its structure unfolds, leading to protein aggregation through intermolecular bonding. This aggregation results in cloudiness or turbidity in tuna slices due to light scattering. The increase in light reflection of muscle surface is related to white appearance [8]. The increases in lightness and yellowness of Asian sea bass slices were caused by protein denaturation when CO_2_ was used as a working gas in modified atmosphere packaging [73].

#### 3.4.2. Mb Derivatives

At the beginning (0 h), metMb, oxyMb, and deoxyMb contents of tuna slices treated with EGCG or quercetin at varying concentrations were in the range of 39.62–40.02%, 34.34–36.59%, and 24.26–26.18%, respectively (Figure 9A–C). Similarly, Chiou et al. [74] documented the same range of metMb in fresh tuna meat. Absorption ability of Mb to visible light is determined by the conjugated double bonds of heme groups and concentration of Mb, which act as pigments [9,12,75]. When comparing the different treatments throughout the storage, the E-200 sample had lower metMb content than the control, E-400, Q-200, and Q-400 samples (*p* < 0.05). This result agreed with in vitro study of metMb solution treated with EGCG (Figure 5A). During the extended storage (72 h), the Q-400 samples had similar metMb content to E-200 (*p* > 0.05) at 72 h. This result was correlated with redness (Figure 8A,B). Notably, there was less metMb content found after 36 h of storage, compared to 24 h of storage in all treated samples. At the same time, a high level of oxyMb content was noticed (Figure 9B). This was due to the reducing activity of endogenous reducing agents such as NADH-dependent MetMb reductase, which was pivotal in reducing oxidized hemeproteins. Cyt-C and Cyt-b5 reductase facilitate metMb reduction, with Cyt-C transferring electrons from NADH nonenzymatically, while Cyt-b5 reductase mediates enzymatic reduction by transferring electrons from NADH [76,77]. Fresh fish have the dominant ferrous state (Mb/Hb-Fe^2+^), contributing to the meat’s bright-red or purple-red color [7]. Nevertheless, the iron atom within the porphyrin ring can undergo oxidation to the ferric state (Mb/Hb-Fe^3+^), resulting in the formation of a brownish color [22].

On the other hand, oxyMb was more strongly stabilized in the E-200 sample (*p* < 0.05) than E-400, Q-200, and Q-400. These findings were opposite to *in vitro* study, where quercetin efficiently increased the oxyMb content, regardless of concentration used. Due to the formation of quinone intermediates from quercetin, the intermediates might undergo interaction with other biomolecules (such as other proteins, lipids, etc.), thus lowering oxyMb stabilizing capability [67,69]. For deoxyMb, its value was decreased after 12 h of storage period. Subsequently, deoxyMb was slightly changed with rising storage time for all the samples (Figure 9C). Moreover, slight change in deoxyMb in treated and untreated samples was considerably important in terms of color stability as deoxyMb undergoes oxidation faster than oxyMb [78].

#### 3.4.3. Lipid Oxidation

After 12 h, the lowest PV was attained for E-200 and E-400 (*p* < 0.05) (Figure 10A). Thereafter, E-400 had a higher PV than the E-200 sample (*p* < 0.05) after 24 h. The Q-200 and Q-400 samples had the highest PV, compared to all treated samples (*p* < 0.05) but there was no difference between Q-400 and E-400 (*p* < 0.05) after 24 and 36 h. EGCG contains 8 hydroxyl groups (-OH) in its structure, whereas quercetin has 5 hydroxyl groups (-OH). Although EGCG was an effective antioxidant in tuna slices, it acted as a prooxidant at high concentrations, leading to the increased formation of hydroperoxide [23]. Nevertheless, the control had the highest increase in PV after 24 h and yielded the maximum PV (*p* < 0.05). However, PV was decreased with increasing storage (36 and 72 h) (*p* < 0.05). The reduction in PV was likely owing to the further decomposition of unstable hydroperoxides. Unsaturated fatty acids were oxidized upon exposure to oxygen, in which hydroperoxides were formed. The unstable hydroperoxide was decomposed to secondary oxidation compounds, which produce undesirable effects such as rancidity or off-odor [18,79].

A continuous upsurge in the TBARS value was noticed in the control (*p* < 0.05) throughout the storage of 72 h (Figure 10B). Hydroperoxides were decomposed into aldehydes at a high extent. After 72 h, the TBARS value of Q-400 was slightly reduced (*p* < 0.05). This could result from the loss of low MW decomposition products [80]. Additionally, aldehyde could form the adduct with other compounds, particularly proteins, leading to a decrease in TBARS [7]. Furthermore, lipid oxidation can darken tuna color by generating melanoidins [81]. Thiansilakul et al. [7] found that metMb and methemoglobin in washed bighead carp mince increased lipid oxidation. Autoxidation of Mb led to the formation of the highly reactive prooxidant, and the loss of heme structure owing to the secondary product (H_2_O_2_), which could induce lipid oxidation. After 72 h, E-200 had the lowest TBARS value (*p* < 0.05). The highest TBARS value was found in the E-400 sample (*p* < 0.05), indicating prooxidant activity at a higher concentration of EGCG. Polyphenols exhibit their antioxidative properties by chelating transition metal ions and scavenging free radicals, thereby delaying meat and fish oxidation [82,83]. The results agreed with the lower redness index (Figure 8B) and higher metMb content in the E-400 sample (Figure 9A). TBARS content was positively correlated with metMb content but negatively related to the redness index (a*/b*) [84]. During the storage, the phenolic compounds might have undergone changes as influenced by several factors such as light, temperature, oxygen, etc. Autoxidation, polymerization, as well as enzymatic modification of the phenolics might have taken place during the storage. Phenolics are structurally unstable due to aromatic rings bearing one or more hydroxyl groups, together with several other substituents [14]. As a result, they can be easily modified by oxidative enzymes present in tuna slices and other factors. The loss of redness of tuna slices might be related to the instability of polyphenols [12]. With the increasing decomposed form of phenolic concentration associated with the extended storage time, the reducing power and scavenging of free radicals became lower. As a result, the ability to maintain the color of the tuna slice became less. Consequently, color changes and lipid oxidation of tuna slices could be enhanced with augmenting storage time. In general, lipid oxidation products are also associated with discoloration [22].

#### 3.4.4. Protein Oxidation

Protein oxidation was determined by measuring the total sulfhydryl and carbonyl contents. At the beginning (0 h), total sulfhydryl and carbonyl content ranged from 5.51 to 5.57 µmole/mg protein and from 1.64 to1.9 µmole/mg protein, respectively (Figure 11A,B). Lipid oxidation products have been shown to induce protein cross-linking and carbonylation reactions [85]. Among the treatments, the E-200-treated sample showed higher total sulfhydryl content (*p* < 0.05) than the E-400- and Q-200-treated samples. A similar trend was observed after 36 h of storage. On the other hand, Q-400 and E-200 had higher values (*p* < 0.05) than E-400 and Q-200 after 24 h of storage. The control had a sharp decrease in SH content throughout the storage period. This indicated that the SH group most likely underwent oxidation to disulfide bonds [86]. The non-helical region of muscle proteins may undergo conformational changes during storage, which could disrupt heme proteins and release free iron. This would cause an acceleration of protein and lipid autoxidation [23,31,87]. At the end of the storage, E-200 had a higher total SH content than others (*p* < 0.05). Phenolic compounds, acting as antioxidants, donate electrons or hydrogen atoms. However, they could have been oxidized to quinones or semi-quinones, thus facilitating the oxidation of SH group to disulfide bonds [14].

Carbonyl compounds could result in oxidative deterioration of side chains of lysine, proline, arginine, and histidine residues [88]. With augmenting storage, the control showed the markedly increased carbonyl content (*p* < 0.05). The lowest carbonyl content was detected in the E-200 sample (*p* < 0.05), compared to other samples. Increases in carbonyl content indicated that tuna meat underwent oxidative deterioration during refrigerated storage, as carbonyls are the primary products of protein autoxidation [89]. Phenolic compounds inhibited protein oxidation by scavenging reactive oxygen radicals by acting similarly to their effect on lipids. Hydroxyl groups in phenolic compounds act as efficient hydrogen donors and interact with protein radicals to form stable compounds, thereby lowering protein oxidation [82,83]. EGCG could bind with the hydrophobic pocket of Mb, mediated by hydrophobic interactions, hydrogen bonds, and van der Waals forces. Hydrogen bonds are particularly significant in this binding process, contributing to the stability of both Mb conformation and the spatial position of EGCG within the hydrophobic pocket [14,90]. Overall, the E-200 sample had the lowest protein oxidation as witnessed by the lowest protein carbonyl content and the most retained SH group content.

#### 3.4.5. Fatty Acid Profile

Fatty acid profiles of fresh tuna slices and those treated with EGCG or quercetin at 200 and 400 mg/kg after 72 h at 4 °C are tabulated in Table 1. Lipids of fresh tuna slices contained 33.06% SFA (saturated fatty acids), 19.35% MUFA (monounsaturated fatty acids), and 47.6% PUFA (poly unsaturated fatty acids). DHA (docosahexaenoic acid) was found to be the most abundant fatty acid (33.83%), followed by palmitic acid (21.34%), and oleic acid (15.11%). EPA (eicosapentaenoic acid) accounted for 5.02%. EPA and DHA are essential PUFAs for human health, being very important for the development and functionality of certain organs [91]. EPA/DHA of the control, E-200, E-400, Q-200, and Q-400 samples were reduced to 5.52/22.94, 5.49/23.7, 4.9/28.67, 7.3/26.51, and 5.1/28.82%, respectively. Other fatty acids were increased as storage time was upsurged. This was related to enhanced lipid oxidation as the storage time of tuna slices was augmented to 72 h as witnessed by the increased TBARS values (Figure 10B). MUFAs and PUFAs undergo oxidation rapidly in the presence of reactive species [22,92]. After 72 h of storage, E-200 showed significantly more PUFAs, compared to control (*p* < 0.05). Additionally, E-200 was observed to have insignificantly higher PUFA content than other treatments with phenolic compounds after 72 h of storage (*p* > 0.05). EGCG at 200 mg/kg was able to scavenge free radicals; however, it became a prooxidant at a higher concentration (400 mg/kg). This result was in tandem with the lower PV and TBARS values of the E-200 sample (Figure 10A,B).

Overall, EGCG, especially at a 200 mg/kg concentration, inhibited the lipid oxidation of tuna slices effectively. Since phenolic compounds do not significantly alter the macronutrient composition of tuna slices, their treatment might have a negligible negative impact on the nutritional value. This antioxidant capability of phenolic compounds could contribute to nutritional integrity by preventing oxidation (protein and fatty acids oxidation) associated with the loss in nutritive value. Furthermore, phenolic compounds include important phytochemicals linked to health benefits such as antioxidant, anti-inflammatory, and anti-cancer capabilities [58]. Thus, incorporating phenolic compounds not only preserves the color of tuna slices but also improves the nutritional quality of the product while offering additional health benefits to consumers, which can be explored further.

## 4. Conclusions

The addition of phenolic compounds to longtail tuna slices was able to maintain the color as well as prevent lipid and protein oxidation, thus retarding the loss in chemical quality. It was found that tuna slices added with EGCG, especially at 200 mg/kg, had a lower formation of metMb and maintained the redness with decreased lipid and protein oxidation, compared to those added with EGCG at high concentrations and other phenolic compounds. Although quercetin enhanced the formation of oxyMb in purified metMb solution, its interaction with other components was enhanced in tuna muscle. The development of brown color in tuna slices added with quercetin predominantly after 12 and 24 h suggested the rejection of quercetin-treated tuna slices for making sashimi products. Furthermore, future research should prioritize the elucidation of the structure in the relationship between plant polyphenols and Mb.

## Figures and Tables

**Figure 1 foods-13-01238-f001:**
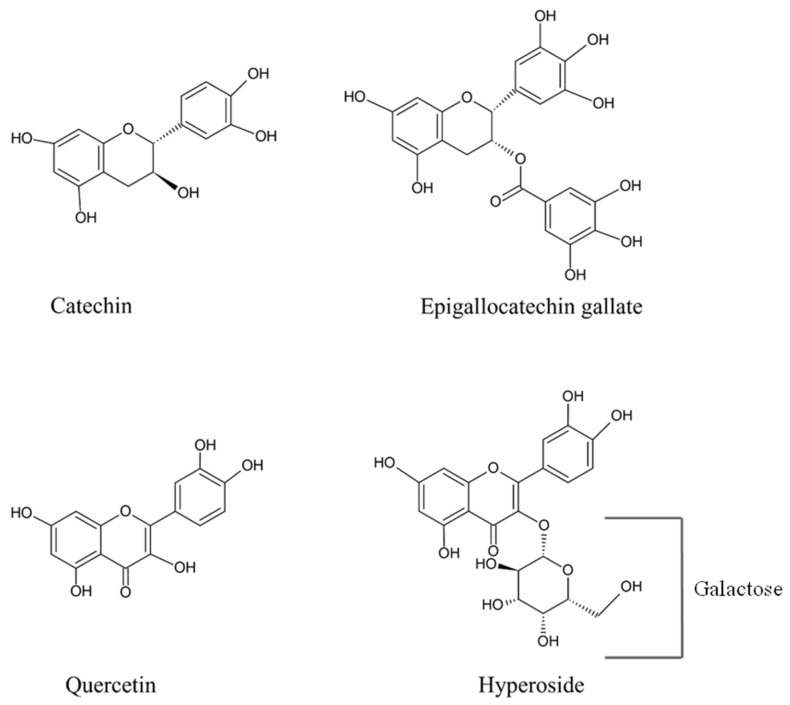
Structure of four phenolic compounds.

**Figure 2 foods-13-01238-f002:**
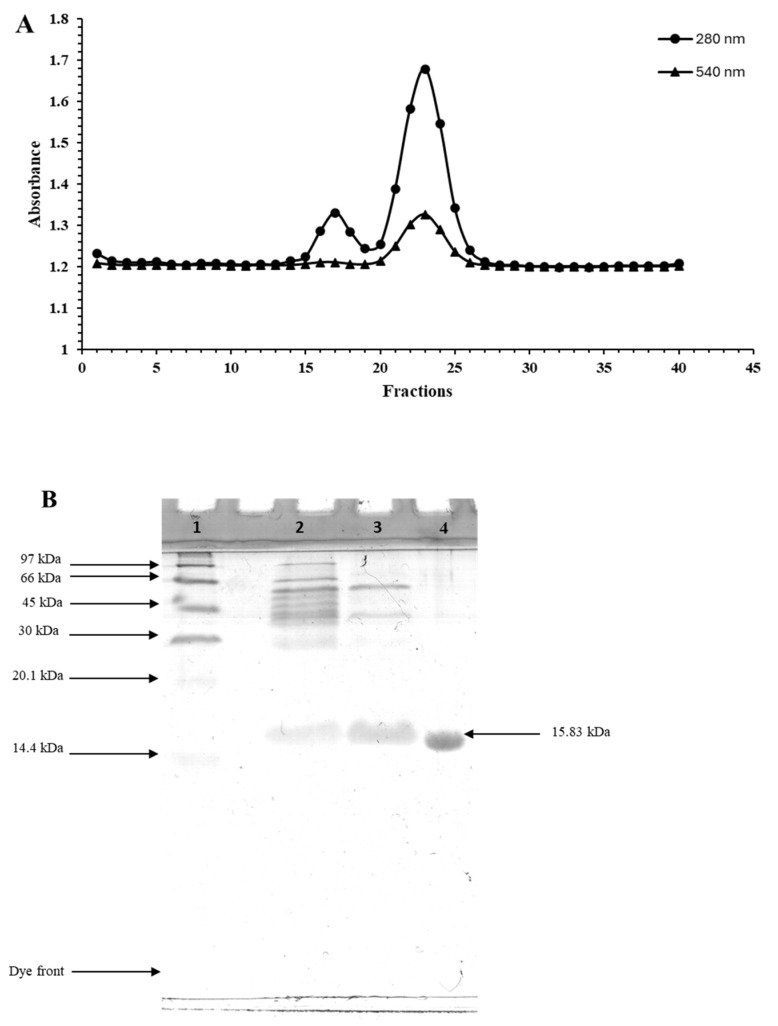
Elution profile of ammonium sulphate fraction (65–100% saturation) containing myoglobin from longtail tuna dark muscle using Sephadex G-50 column (**A**) and SDS–PAGE pattern of the crude myoglobin extract and different fractions (**B**). Lane 1: low molecular weight marker; lane 2: crude myoglobin extract; lane 3: ammonium sulphate fraction (65–100% saturation; AS) after dialysis; lane 4: Sephadex G-50 fraction (pooled fraction Nos. 20–25).

**Figure 3 foods-13-01238-f003:**
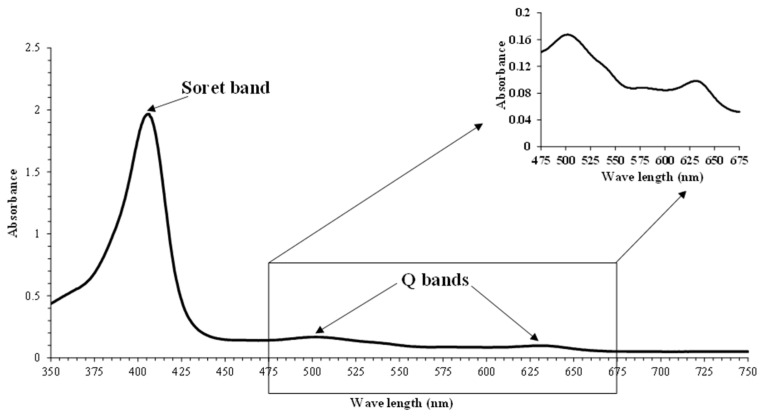
Absorption spectrum of purified myoglobin from longtail tuna dark muscle.

**Figure 4 foods-13-01238-f004:**
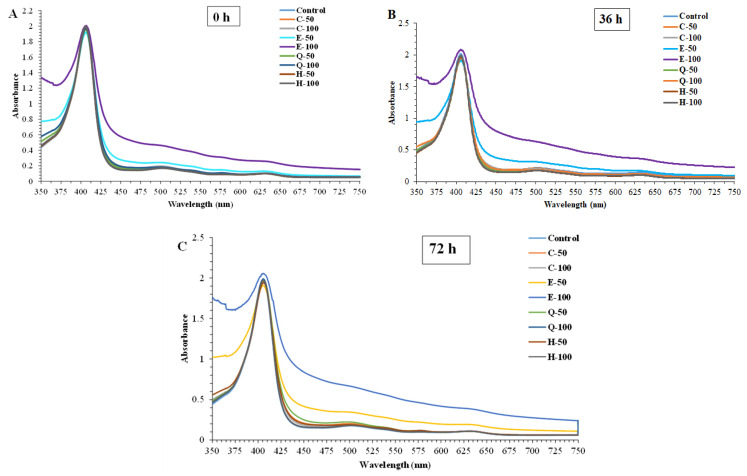
Effect of some phenolic compounds at different concentrations on absorption spectra of purified myoglobin from longtail tuna dark muscle as a function of time including 0 (**A**), 36 (**B**), and 72 h (**C**). Control: without phenolic treatments; C-50: 50 ppm catechin; C-100: 100 ppm catechin; E-50: 50 ppm EGCG; E-100: 100 ppm EGCG; Q-50: 50 ppm quercetin; Q-100: 100 ppm quercetin; H-50: 50 ppm hyperoside; H-100: 100 ppm hyperoside.

**Figure 5 foods-13-01238-f005:**
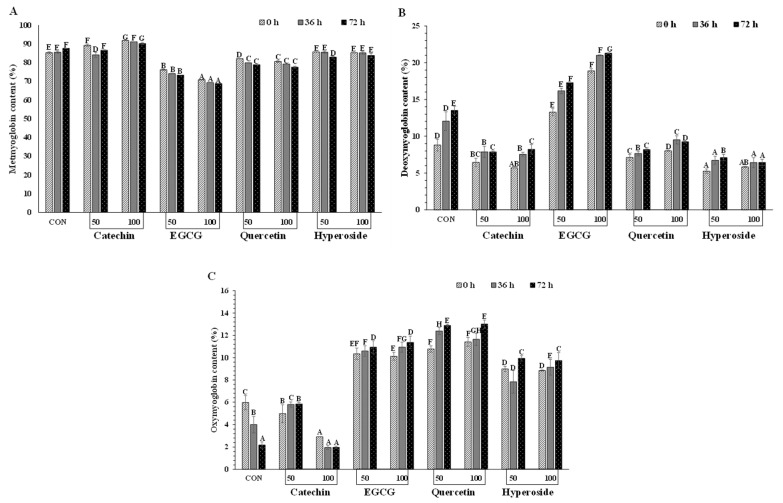
Effect of some phenolic compounds at different concentrations (50 and 100 ppm) on different forms of myoglobin: metmyoglobin (**A**), deoxymyoglobin (**B**), and oxymyoglobin (**C**) in metmyoglobin solution from longtail tuna dark muscle as a function of time. Different uppercase letters on the bar denote significant differences (*p* < 0.05). Bars represent the standard deviations (*n* = 3). Numbers 50 and 100 indicate the concentrations of phenolics in ppm.

**Figure 6 foods-13-01238-f006:**
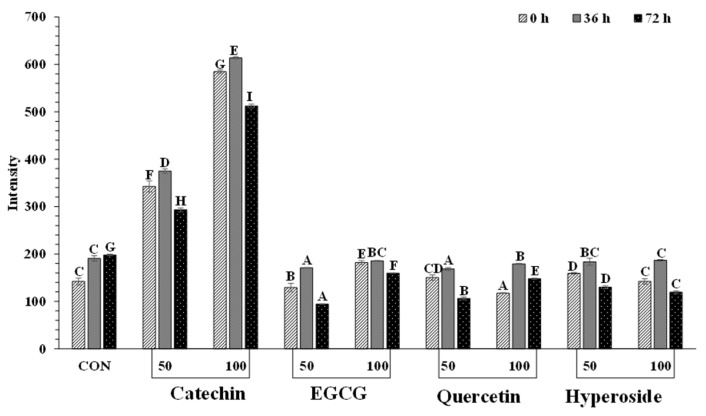
Effect of some phenolic compounds at different concentrations (50 and 100 ppm) on tryptophan fluorescent intensity in metmyoglobin solution from longtail tuna dark muscle as a function of time. Different uppercase letters on the bar denote significant differences (*p* < 0.05). Bars represent the standard deviations (*n* = 3). Numbers 50 and 100 indicate the concentrations of phenolics in ppm.

**Figure 7 foods-13-01238-f007:**
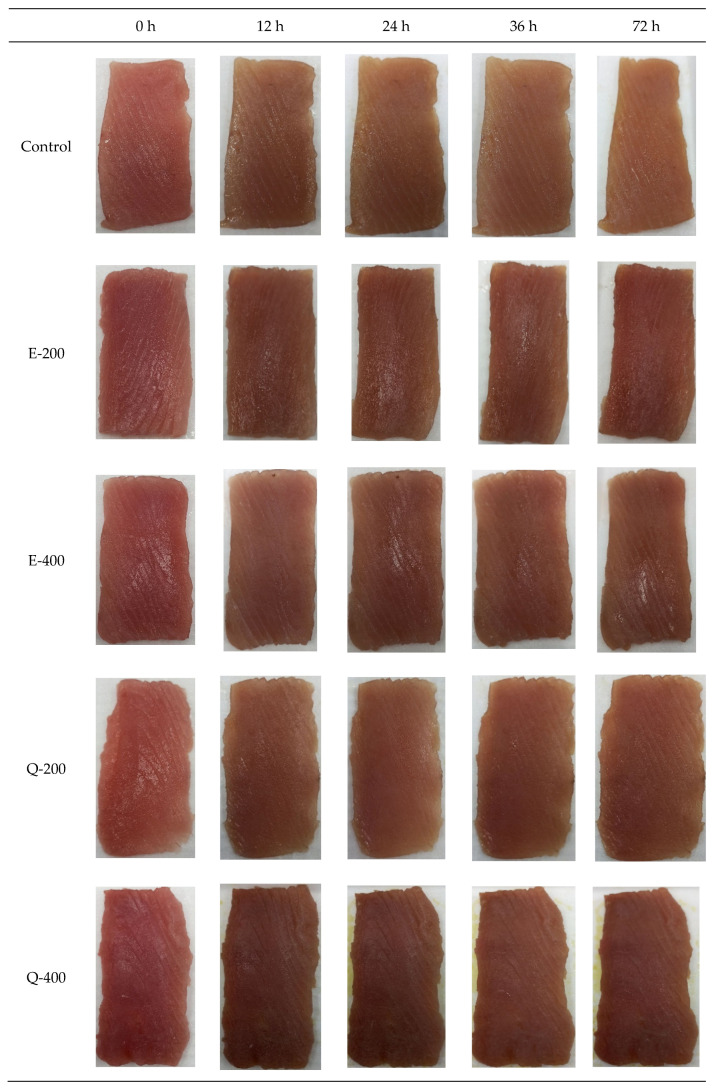
Photographs of longtail tuna slices treated with EGCG or quercetin at different concentrations during storage of 72 h at 4 °C. Control: without phenolic compounds; E-200: slices treated with 200 ppm EGCG; E-400: slices treated with 400 ppm EGCG: Q-200: slices treated with 200 ppm quercetin; Q-400: slices treated with 400 ppm quercetin.

**Figure 8 foods-13-01238-f008:**
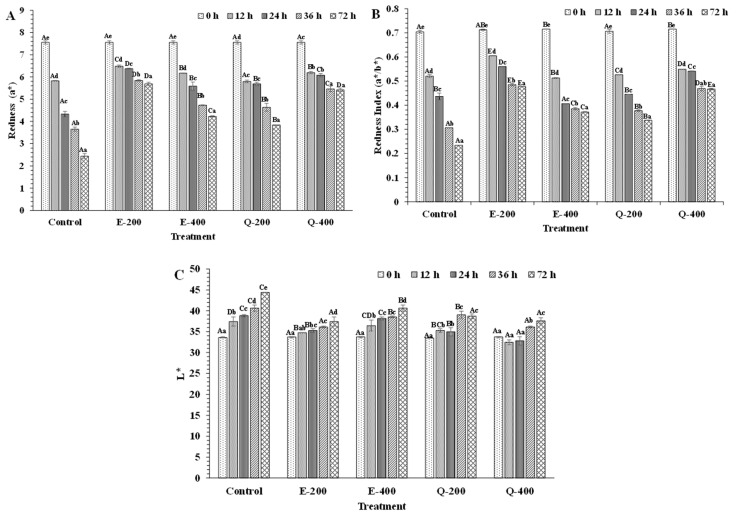
Changes in redness (a* value) (**A**), redness index (a*/b*) (**B**) and lightness (L* value) (**C**) of longtail tuna slices treated with EGCG and quercetin at different concentrations during storage of 72 h at 4 °C. Different uppercase letters on the bars between different samples denote significant differences (*p* < 0.05). Different lowercase letters on the bars within the same treatment denote significant differences (*p* < 0.05). Bars represent the standard deviations (*n* = 3). Caption: See Figure 7.

**Figure 9 foods-13-01238-f009:**
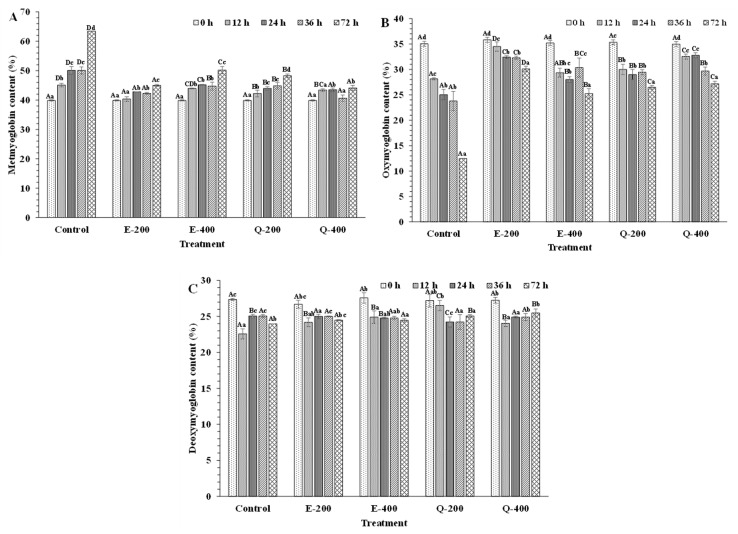
Changes in different forms of myoglobin: metmyoglobin (**A**), oxymyoglobin (**B**), and deoxymyoglobin (**C**) contents of fresh longtail tuna slices treated without and with EGCG and quercetin at different concentrations during storage at 4 °C. Different uppercase letters on the bars between different samples denote significant differences (*p* < 0.05). Different lowercase letters on the bars within the same treatment denote significant differences (*p* < 0.05). Bars represent the standard deviations (*n* = 3). Caption: See Figure 7.

**Figure 10 foods-13-01238-f010:**
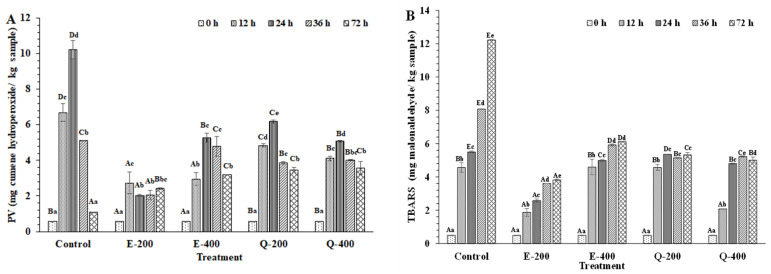
Changes in peroxide value (PV) (**A**) and thiobarbituric acid reactive substances (TBARS) (**B**) of longtail tuna slices treated with EGCG and quercetin at different concentrations during storage of 72 h at 4 °C. Different uppercase letters on the bars between different samples denote significant differences (*p* < 0.05). Different lowercase letters on the bars within the same treatment denote significant differences (*p* < 0.05). Bars represent the standard deviations (*n* = 3). Caption: See Figure 7.

**Figure 11 foods-13-01238-f011:**
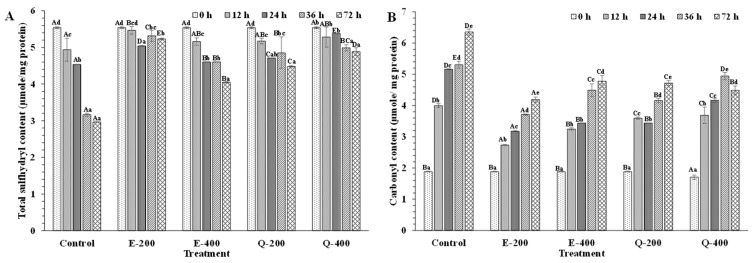
Changes in total sulfhydryl (**A**) and carbonyl content (**B**) of longtail tuna slices treated with EGCG and quercetin at different concentrations during storage of 72 h at 4 °C. Different uppercase letters on the bars between different samples denote significant differences (*p* < 0.05). Different lowercase letters on the bars within the same treatment denote significant differences (*p* < 0.05). Bars represent the standard deviations (*n* = 3). Caption: See Figure 7.

**Table 1 foods-13-01238-t001:** Changes in fatty acid profile of fresh tuna slices and those without and with EGCG and quercetin treatment at different concentrations after 72 h of storage at 4 °C.

Fatty Acids	Formula	Fresh	Control	E-200	E-400	Q-200	Q-400
Lauric acid	C12:0	0.13 ± 0.02 ^b^	0.08 ± 0.02 ^cd^	0.06 ± 0 ^d^	0.59 ± 0.01 ^a^	0.09 ± 0.02 ^c^	0.09 ± 0.01 ^c^
Myristic acid	C14:0	1.58 ± 0.03 ^d^	2.03 ± 0.17 ^c^	2.91 ± 0.01 ^a^	2.16 ± 0.02 ^c^	2.55 ± 0.28 ^b^	2.27 ± 0.15 ^c^
Pentadecanoic acid	C15:0	0.51 ± 0.1 ^c^	0.68 ± 0.07 ^b^	0.87 ± 0 ^a^	0.62 ± 0.01 ^bc^	0.92 ± 0.1 ^a^	0.67 ± 0.1 ^b^
Palmitic acid	C16:0	21.34 ± 0.15 ^c^	25.61 ± 0.32 ^a^	26.45 ± 0.11 ^a^	23.18 ± 0.05 ^bc^	26.17 ± 0.92 ^a^	23.57 ± 1 ^bc^
Heptadecanoic acid	C17:0	0.82 ± 0 ^c^	0.97 ± 0.09 ^b^	1.11 ± 0 ^b^	0.81 ± 0.07 ^c^	1.46 ± 0.15 ^a^	1.52 ± 0.05 ^a^
Stearic acid	C18:0	7.91 ± 0.05 ^bc^	9.16 ± 0.86 ^b^	0.09 ± 0.01 ^bc^	7.62 ± 0.06 ^c^	11.67 ± 1.45 ^a^	8.19 ± 0.1 ^bc^
Arachidic acid	C20:0	0.23 ± 0.09 ^b^	0.31 ± 0.06 ^b^	0.97 ± 0.56 ^a^	0.33 ± 0.01 ^b^	0.58 ± 0.06 ^ab^	0.26 ± 0.01 ^b^
Docosanoic acid	C22:0	0.26 ± 0.09 ^bc^	0.14 ± 0.02 ^cd^	0.29 ± 0.08 ^ab^	0.23 ± 0.09 ^bcd^	0.4 ± 0.04 ^a^	0.13 ± 0.02 ^d^
Lignoceric acid	C24:0	0.28 ± 0.03 ^bcd^	0.29 ± 0.03 ^bc^	0.25 ± 0 ^cd^	0.24 ± 0.01 ^d^	0.39 ± 0.03 ^a^	0.3 ± 0.02 ^b^
SFA		33.06 ± 0.51 ^d^	39.27 ± 3.36 ^bc^	41 ± 0.50 ^ab^	35.78 ± 0.02 ^c^	44.23 ± 5.041 ^a^	37 ± 1.23 ^bc^
Palmitoleic acid	C16:1	3.41 ± 0 ^d^	4.88 ± 0.45 ^a^	4.61 ± 0.04 ^ab^	4.48 ± 0.07 ^ab^	3.83 ± 0.49 ^cd^	4.1 ± 0.15 ^bc^
Cis-10-Heptadecenoic acid	C17:1	0.09 ± 0.03 ^b^	0.09 ± 0.02 ^b^	0.13 ± 0 ^a^	0.15 ± 0.01 ^a^	0.12 ± 0.02 ^ab^	0.05 ± 0 ^c^
Oleic acid	C18:1 n9c	15.11 ± 0.07 ^ab^	17.86 ± 1.7 ^a^	9.15 ± 0.13 ^c^	17 ± 0.14 ^a^	9.09 ± 1.03 ^c^	15.9 ± 0.5 ^ab^
Elaidic acid	C18:1n9t	0.12 ± 0.01 ^ab^	0.08 ± 0.04 ^bcd^	0.07 ± 0.02 ^cd^	0.05 ± 0.03 ^d^	0.11 ± 0.01 ^bc^	0.15 ± 0 ^a^
Cis-11-Eicosenoic acid	C120:1	0.24 ± 0.18 ^a^	0.1 ± 0.03 ^a^	0.67 ± 0.61 ^a^	0.45 ± 0.4 ^a^	0.11 ± 0 ^a^	0.64 ± 0.05 ^a^
Erucic acid	C22:1 n9	0.05 ± 0 ^a^	0.19 ± 0.14 ^a^	0.08 ± 0.01 ^a^	0.18 ± 0.12 ^a^	0.12 ± 0.08 ^a^	0.17 ± 0.02 ^a^
Nervonic acid	C24:1	0.33 ± 0.04 ^bc^	0.34 ± 0.05 ^bc^	0.44 ± 0.02 ^a^	0.3 ± 0 ^c^	0.39 ± 0.06 ^ab^	0.35 ± 0.02 ^bc^
MUFA		19.35 ± 0.18 ^abc^	23.54 ± 1.99 ^a^	15.15 ± 9.82 ^bc^	22.61 ± 0.42 ^ab^	13.77 ± 1.52 ^c^	21.36 ± 0.40 ^abc^
Linoleic acid	C18:2n6c	1.86 ± 0.03 ^b^	1.74 ± 0.18 ^b^	9.92 ± 0.42 ^a^	2.16 ± 0.62 ^b^	1.61 ± 0.25 ^b^	1.72 ± 0.05 ^b^
γ-Linolenic acid	C18:3n6	0.05 ± 0.03 ^b^	0.11 ± 0.01 ^b^	0.25 ± 0.15 ^ab^	0.45 ± 0.4 ^a^	0.1 ± 0.01 ^b^	0.05 ± 0 ^b^
α-Linolenic acid (ALA)	C18:3n3	0.15 ± 0.09 ^a^	0.46 ± 0.03 ^a^	0.47 ± 0.42 ^a^	0.22 ± 0.15 ^a^	0.53 ± 0.35 ^a^	0.29 ± 0.01 ^a^
Eicosadienoic acid	C20:2	0.32 ± 0.18 ^c^	0.36 ± 0.1 ^c^	0.39 ± 0.06 ^c^	0.32 ± 0.09 ^c^	0.99 ± 0.16 ^a^	0.75 ± 0.1 ^b^
Eicosatetraenoic acid	C20:4	6.37 ± 0.05 ^a^	6.04 ± 0.58 ^ab^	3.63 ± 0.01 ^d^	4.88 ± 0.07 ^c^	4.94 ± 0.66 ^c^	5.44 ± 0.5 ^bc^
Eicosapentaenoic acid (EPA)	C20:5n3	5.02 ± 0.02 ^b^	5.52 ± 0.48 ^b^	5.49 ± 0.03 ^b^	4.9 ± 0.22 ^b^	7.3 ± 0.88 ^a^	5.11 ± 0 ^b^
Docosahexaenoic acid (DHA)	C22:6n3	33.83 ± 0.09 ^a^	22.94 ± 0.81 ^d^	23.7 ± 0.31 ^d^	28.67 ± 0.22 ^ab^	26.51 ± 0.87 ^ac^	28.82 ± 0.84 ^ab^
PUFA		47.6 ± 0.32 ^a^	37.17 ± 5.63 ^b^	43.85 ± 9.32 ^ab^	41.6 ± 0.44 ^ab^	41.98 ± 6.56 ^ab^	42.18 ± 0.29 ^ab^

Control: without phenolic compounds; E-200: slices treated with 200 ppm EGCG; E-400: slices treated with 400 ppm EGCG; Q-200: slices treated with 200 ppm quercetin; Q-400: slices treated with 400 ppm quercetin. Values represent mean and standard deviation (*n* = 3). Different lowercase superscripts in the same row indicate significant differences among treatments with storage time (*p* < 0.05).

## Data Availability

The original contributions presented in the study are included in the article, further inquiries can be directed to the corresponding author.

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
