# Peer review of "Effects of Different Phenolic Compounds on the Redox State of Myoglobin and Prevention of Discoloration, Lipid and Protein Oxidation of Refrigerated Longtail Tuna (Thunnus tonggol) Slices"

_foods, 2024, doi:10.3390/foods13081238_

Round 1
Reviewer 1 Report
Comments and Suggestions for Authors
-
Here are some of my questions about this manuscript.
- Phenolic Compounds: How do different phenolic compounds specifically affect the redox state of myoglobin in longtail tuna?
- Concentration Effects: What is the impact of varying concentrations of EGCG and quercetin on the quality of tuna slices during refrigeration?
- Storage Duration: How does the duration of storage influence the effectiveness of phenolic compounds in preserving the color and quality of tuna slices?
- Protein Interactions: In what ways do phenolic compounds interact with myoglobin and other proteins in tuna to affect color stability?
- Oxidation Mechanisms: Can the mechanisms by which phenolic compounds prevent oxidation be further elucidated?
- Comparative Analysis: How do the effects of phenolic compounds on longtail tuna compare to their effects on other types of fish or meat products?
- Sensory Evaluation: Were any sensory evaluations conducted to assess consumer perception of color and taste in treated tuna slices?
- Nutritional Impact: Does the treatment with phenolic compounds have any impact on the nutritional value of the tuna slices?
- Shelf Life Extension: What is the maximum shelf life extension achievable with phenolic compound treatments, and how does it compare to other preservation methods?
- Commercial Viability: How commercially viable are the phenolic compound treatments for longtail tuna in terms of cost and application in large-scale processing?
Author Response
Reviewer: 1
Comments to the author
Here are some of my questions about this manuscript.
*****Thank you so much for the invaluable suggestions and insightful comments. All queries have been responded and the corrections have been done as highlighted in yellow color.
Phenolic Compounds: How do different phenolic compounds specifically affect the redox state of myoglobin in longtail tuna?
***** Depending on the structure, functional groups, and concentration, phenolic compounds can affect the redox state of myoglobin through various mechanisms. Most phenolic compounds possess reducing power and are able to donate electrons to the central iron of porphyrin ring in myoglobin molecule, thus facilitating the reduction of Fe³⁺ to Fe²⁺. Additionally, they can indirectly safeguard myoglobin from oxidative damage by scavenging reactive oxygen species (ROS), which could induce the oxidation of myoglobin [22]. These processes highlight the importance of phenolic compounds in determining the redox state of myoglobin in longtail tuna.
Reference
Dragoev, S.G. Lipid Peroxidation in Muscle Foods: Impact on Quality, Safety and Human Health. Foods 2024, 13, 797.
For better understanding, the explanation along with the relevant references has been given in the text. Please see lines 80-84.
Concentration Effects: What is the impact of varying concentrations of EGCG and quercetin on the quality of tuna slices during refrigeration?
***** As mentioned in the above responses, phenolic compounds have capability of reducing Fe3+ to Fe2+, thus converting the oxidized form to reduced form. Since phenolic compounds, especially EGCG, at higher amount more likely provide higher reducing power, it should reduce the met-Mb to oxy-Mb in a dose dependent manner. However, the excessive amount of phenolic compounds can act as prooxidants, thus causing the discoloration.
From our study, EGCG at 200 mg/kg effectively scavenged free radicals in tuna slices. However, it became pro-oxidant at higher concentrations (400 mg/kg). At higher concentration, protein-EGCG interaction might cause rapid discoloration of the tuna slices. Furthermore, in the presence of hydrogen peroxide, generated from the auto-oxidation of myoglobin (Mb) and peroxidase activity, the A ring of quercetin could be oxidized to o-quinone. This o-quinone further converted to a quinone methide isomer, which was more electrophilic and acted as a pro-oxidant, leading to the darkening of tuna slices.
These explanations had already given in line 442 – 449, 433 – 436.
Storage Duration: How does the duration of storage influence the effectiveness of phenolic compounds in preserving the color and quality of tuna slices?
***** The storage duration significantly impacts the effectiveness of phenolic compounds, such as EGCG and quercetin, in preserving the color and quality of tuna slices. Most phenolic compounds undergo decomposition over time, especially when exposed to light, heat, and oxygen. These factors influence the effectiveness of phenolic compounds with increasing storage time.
Initially, the phenolic compounds significantly reduced oxidation by scavenging free radicals and reactive oxygen species (ROS), thereby preserving the color of tuna slices. However, during storage, phenolic compounds undergo degradation due to heat, light, oxygen, and other oxidizing agents. The primary function of phenolic compounds in preserving tuna is their ability to provide the electron or scavenge free radicals. With increasing decomposed form of phenolic concentration associated with the extended storage time, the reducing powder and scavenging of free radicals became lower. As a result, the ability in maintaining the color of tuna slice was less. Consequently, color changes and lipid oxidation of tuna slices could be enhanced by augmenting storage time. Lipid oxidation products are associated with discoloration [22].
For better understanding, the explanation along with the relevant references has been given in the text. Please see lines 560-565.
Protein Interactions: In what ways do phenolic compounds interact with myoglobin and other proteins in tuna to affect color stability?
***** Phenolic compounds can interact with globin, a protein in tuna slice in several ways, particularly through H-bonding or hydrophobic interaction. These interactions significantly affect the color stability of tuna slices. Via the formation of weak bonds with specific amino acid residues in Mb, phenolic compounds can help to maintain its three-dimensional structure. This potentially makes Mb less susceptible to oxidation, which can cause discoloration. However, the degree of interaction more likely depends on the type and concentration of the phenolic compound used [7]. This explanation had already given in lines 372-376.
Reference
Thiansilakul, Y.; Benjakul, S.; Park, S.Y.; Richards, M.P. Characteristics of Myoglobin and Haemoglobin-mediated Lipid Oxidation in Washed Mince from Bighead Carp (Hypophthalmichthys nobilis). Food Chemistry 2012, 132, 892–900.
Oxidation Mechanisms: Can the mechanisms by which phenolic compounds prevent oxidation be further elucidated?
***** There are several mechanisms involved by phenolic compounds to prevent oxidation: Firstly, they scavenge free radicals generated during oxidation, stabilizing them with phenolic hydroxyl groups and terminating the radical chain reaction. Furthermore, they chelate pro-oxidative metal ions like iron and copper, inhibiting oxidation processes catalyzed by these metals.
Those mechanisms have been included in lines 67-73.
Comparative Analysis: How do the effects of phenolic compounds on longtail tuna compare to their effects on other types of fish or meat products?
***** Different fish species have variations in myoglobin content. It may have the difference in globin structure. Those factors could determine the efficacy of the phenolic compounds in maintaining color of varying fish meat in different ways. Additionally, chemical compositions in the muscle, especially lipid and polyunsaturated fatty acid content, associated with their lipid oxidation products, could affect the rate of Mb oxidation at different degrees. As a consequence, the efficiency of phenolic compound in preventing discoloration of fish slices can be varied.
Those explanations have already been added in line 414-421.
Sensory Evaluation: Were any sensory evaluations conducted to assess consumer perception of color and taste in treated tuna slices?
***** Thank you for your valuable comment. We did not conduct sensory evaluation to assess consumer perception of color and taste of treated tuna slices in this study. In the present study, authors try to understand the role of phenolic compound in reducing the met-Mb to oxy-Mb, a desirable form. Also, efficacy of various phenolic compound on changes in met-Mb and appearance of tuna slice was also investigated.
However, we do agree to conduct the sensory evaluation in the sample in our future work, which can provide the insightful information on color and taste of treated tuna slices. Your insightful suggestion is taken into consideration for our future work. Thank you so much.
Nutritional Impact: Does the treatment with phenolic compounds have any impact on the nutritional value of the tuna slices?
***** Since phenolic compounds do not significantly alter the macronutrient composition of tuna slices, their treatment may have a negligible negative impact on the nutritional value [1]. In fact, phenolic compounds possess antioxidant properties that can preserve certain vitamins, like vitamins A and E, which are susceptible to oxidation during storage. This antioxidant capability of phenolic compounds contributes to nutritional integrity by preventing oxidation (protein and fatty acids oxidation) and undesirable discoloration. Furthermore, phenolic compounds include important phytochemicals linked to health benefits such as antioxidant, anti-inflammatory, and anti-cancer capabilities [2].
Reference
- Singh, A.; Benjakul, S.; Zhou, P.; Zhang, B.; Deng, S. Effect of Squid Pen Chitooligosaccharide and Epigallocatechin Gallate on Discoloration and Shelf-Life of Yellowfin Tuna Slices during Refrigerated Storage. Food Chem 2021, 351, 129296.
- Zehiroglu, C.; Ozturk Sarikaya, S.B. The Importance of Antioxidants and Place in Today’s Scientific and Technological Studies. J Food Sci Technol 2019, 56, 4757–4774, doi:10.1007/s13197-019-03952-x.
Shelf Life Extension: What is the maximum shelf life extension achievable with phenolic compound treatments, and how does it compare to other preservation methods?
***** Authors did not study the self-life of tuna slice in this study. The period of 3 days mimics the maximal storage time of sashimi from tuna since discoloration can take place rapidly for tuna slices. It has been known that bacterial growth can induce the change of color of tuna slice. To avoid such an interfering effect caused by microbial growth, NaN3 as an antimicrobial agent, was added to inactivate the microorganism. Please see line 180
For the shelf life study, it will be another study in the near future. Based on our previous work or literatures, phenolic compound can extend the shelf life of fish slices by several days compared to untreated slices via maintaining the quality attributes. Since color is a crucial index used for assessing freshness and acceptance of fresh tuna for sushi and sashimi products, we focused on the color mediated by the changes in Mb chemistry in the present study.
Commercial Viability: How commercially viable are the phenolic compound treatments for longtail tuna in terms of cost and application in large-scale processing?
***** The commercial viability of applying phenolic compound treatments to longtail tuna depends on several factors in large-scale processing. The cost of phenolic compounds depends on the source (natural and synthetic), concentration required, and process integration. For large-scale, cost-effectiveness developing a standardized protocol for different types of phenolics and ensuring compliance with food safety regulations as a food additive are crucial. Advancements in extracting phenolics from cost-effective sources and optimizing application techniques hold the promise for making this technology more commercially viable in the future.
Nowadays, single phenolic compounds are commercially available with the affordable price. Thus, it has high feasibility to use the polyphenolic compounds as the natural additive to replace synthetic counterpart.

Reviewer 2 Report
Comments and Suggestions for Authors
Review foods-2956886_05.04.2024
Comments and Suggestions for Authors
The manuscript entitled Effects of phenolic compounds on characteristics of myoglobin, discoloration, lipid and protein oxidation of refrigerated longtail tuna (Thunnus tonggol) is interesting. It is relatively well structured. It is expertly written. The English language is clear. The expression of the authors is correct from a scientific point of view - without unnecessary speculations and manipulations. The conducted research is definitely of interest to the specialized scientific community, both in practical and fundamental-theoretical aspects. The analysis methods used are modern and well selected. The experimental setup is correct. Statistical analysis is adequate.
A huge amount of data has been obtained and presented in an adequate manner.
All this shows the satisfactory nature of the manuscript and its novelty and high scientific value.
Improvement opportunities
Title, Objectives and Structure of the Article
I would recommend that the authors reformulate the objectives of their study more clearly. At the end of the Introduction section, the aim of the study presented on lines 85-86 is too short and, in my opinion, does not cover all the analysis and results that the paper presents. The purpose of the study formulated in this way is too limited and, in my opinion, does not cover the full volume of new knowledge obtained.
The purpose of the study is not defined at the beginning of the abstract too, which makes it difficult for the readers.
The title of the manuscript is an emanation of the results obtained and the conclusions drawn. Judging by the reported results, I recommend that the authors discuss a revision of the title in line with the new revision of the purpose of the experiment.
Abstract
The summary is informative and well structured. I ask the authors to discuss whether it is not a good idea to put the objectives of their study as the first sentence and to emphasize the conclusions drawn as the last.
1. Introduction
The introduction is concise, but sufficiently informative and well introduces the reader to the study of a scientific and practical problem. The literature used in the references is satisfactory. However, I ask the authors to discuss the inclusion in the Introduction and in the discussion of the obtained results of some publications that are directly relevant and tangential to the discussed problems. For example:
- Lipid peroxidation in muscle foods: Impact on quality, safety and human health. Foods, 13(5), Article ID 797. https://doi.org/10.3390/foods13050797;
I have some notes to fix technical inconsistencies in the writing of the manuscript, e.g.
Line 201 - It says: "...of 13,600 M−1cm−1." I ask the authors to uppercase -1 and let the dimension be represented as: 13,600 M−1cm−1.
Line 229 – it is written: "(fraction Nos. 20–25)" to put a hyphen - "(fraction Nos. 20-25)"
Line 253 - it is written: "...350–750 nm" to insert a hyphen - "...350-750 nm"
Line 258 – it is written: of 450–750 nm, the" to put a hyphen - "of 450-750 nm, the"
Line 463 - "... when CO2 was used..." - the carbon dioxide formula to be correct - CO2
Line 491 - "... (Mb/Hb-Fe2+)," to become "...(Mb/Hb-Fe2+),"
Line 493 - "...(Mb/Hb-Fe3+)," to become "...(Mb/Hb-Fe3+),"
2. Materials and Methods
In section 2.4. Effect of phenolic compounds on the proportion of Mb forms in metMb solution
Lines 142 - 144 – Give a little more information why exactly these phenolic antioxidant preparations were chosen? A similar justification is also missing in the Introduction section!
3. Results and Discussion
In Table 1 on pages 18 and 19 - to present the confidence intervals of variation as mean ± standard deviation or mean ± standard error of the means. To indicate the probability (p) . Put letters indicating which means are statistically distinguishable and which are not? As done in the figures above.
5. Conclusions
Line 607 – number 5 is written Conclusions. In my opinion, it should be number 4. Conclusions.
I recommend that the authors consider revising the Conclusions in light of possible changes in the wording of the study objectives and the significant volume of results presented.
It is recommended that this section begin with a single sentence that indicates to the reader whether the null hypothesis of the study is confirmed or rejected. In the next two - three sentences should be formulated in the passive tense according to the important conclusions for fundamental science and for practice. E.g.: It was found that...
Based on the above, I propose minor revision.
Author Response
Reviewer: 2
Comments and Suggestions for Authors
The manuscript entitled Effects of phenolic compounds on characteristics of myoglobin, discoloration, lipid and protein oxidation of refrigerated longtail tuna (Thunnus tonggol) is interesting. It is relatively well structured. It is expertly written. The English language is clear. The expression of the authors is correct from a scientific point of view - without unnecessary speculations and manipulations. The conducted research is definitely of interest to the specialized scientific community, both in practical and fundamental-theoretical aspects. The analysis methods used are modern and well selected. The experimental setup is correct. Statistical analysis is adequate.
A huge amount of data has been obtained and presented in an adequate manner.
All this shows the satisfactory nature of the manuscript and its novelty and high scientific value.
*****Thank you so much for your invaluable time spent on our manuscript. All queries have been responded and the corrections have been made following the reviewer’s suggestion as highlighted in green color.
Improvement opportunities
Title, Objectives and Structure of the Article
I would recommend that the authors reformulate the objectives of their study more clearly. At the end of the Introduction section, the aim of the study presented on lines 85-86 is too short and, in my opinion, does not cover all the analysis and results that the paper presents. The purpose of the study formulated in this way is too limited and, in my opinion, does not cover the full volume of new knowledge obtained.
***** Thank you for the insightful suggestion regarding the title. The objectives of our study were reformulated to cover the major analyses for better clarity. Please see lines 92-97
The purpose of the study is not defined at the beginning of the abstract too, which makes it difficult for the readers.
***** Thanks for the insightful comment. The purpose of the study has been defined in the abstract as suggested by the reviewer. Please see lines 14 -16.
The title of the manuscript is an emanation of the results obtained and the conclusions drawn. Judging by the reported results, I recommend that the authors discuss a revision of the title in line with the new revision of the purpose of the experiment.
***** Thank you for your insightful suggestions. The title of our study has been revised. Please see lines 2-4.
Abstract
The summary is informative and well structured. I ask the authors to discuss whether it is not a good idea to put the objectives of their study as the first sentence and to emphasize the conclusions drawn as the last.
*****Thank you so much for your valuable comment. The abstract was restructured. The objective has been provided at the beginning, while the conclusion has been already provided at the end. Please see lines 14-16 and 26-27.
- Introduction
The introduction is concise, but sufficiently informative and well introduces the reader to the study of a scientific and practical problem. The literature used in the references is satisfactory. However, I ask the authors to discuss the inclusion in the Introduction and in the discussion of the obtained results of some publications that are directly relevant and tangential to the discussed problems. For example:
- Lipid peroxidation in muscle foods: Impact on quality, safety and human health. Foods, 13(5), Article ID 797. https://doi.org/10.3390/foods13050797;
***** Thank you very much for your suggestions on the useful reference. The addition of suggestion reference has been done. Please see the lines 64-67, 80-84, 511-513 and 622-623.
I have some notes to fix technical inconsistencies in the writing of the manuscript, e.g.
Line 201 - It says: "...of 13,600 M−1cm−1." I ask the authors to uppercase -1 and let the dimension be represented as: 13,600 M−1cm−1.
***** Sorry for typing error. “13,600 M−1cm−1” has been changed to ‘13,600 M−1cm−1’. Please see line 212
Line 229 – it is written: "(fraction Nos. 20–25)" to put a hyphen - "(fraction Nos. 20-25)"
*****The words "(fraction Nos. 20–25)" has been changed to ‘(fraction Nos. 20-25)’. Please see line 240
Line 253 - it is written: "...350–750 nm" to insert a hyphen - "...350-750 nm"
Line 258 – it is written: of 450–750 nm, the" to put a hyphen - "of 450-750 nm, the"
*****A hyphen has been used for 350-750 and 450-750 nm. Please see the line 264 and 269.
Line 463 - "... when CO2 was used..." - the carbon dioxide formula to be correct - CO2
*****Sorry for the mistake. The subscript has been used for CO2. Please see the line 482.
Line 491 - "... (Mb/Hb-Fe2+)," to become "...(Mb/Hb-Fe2+),"
Line 493 - "...(Mb/Hb-Fe3+)," to become "...(Mb/Hb-Fe3+),"
*****Corrections have been made. Please see the lines 510, 512.
- Materials and Methods
In section 2.4. Effect of phenolic compounds on the proportion of Mb forms in metMb solution
Lines 142 - 144 – Give a little more information why exactly these phenolic antioxidant preparations were chosen? A similar justification is also missing in the Introduction section!
*****Thank you so much for valuable comment.
Synthetic antioxidants can have toxicity concerns. Therefore, natural phenolic compounds from foods or plants as functional food ingredients have been explored. These compounds are generally considered safe and have been part of human diets for centuries. The antioxidant mechanisms of these phenolic compounds vary significantly from one to another.
Please see the lines 83-84.
- Results and Discussion
In Table 1 on pages 18 and 19 - to present the confidence intervals of variation as mean ± standard deviation or mean ± standard error of the means. To indicate the probability (p). Put letters indicating which means are statistically distinguishable and which are not? As done in the figures above.
*****Statistical analysis has been performed and the letters indicating the significant differences have been provided. See Table 1. The text regarding the comparison has been cross-checked based on the statistical analysis. Thank you so much.
- Conclusions
Line 607 – number 5 is written Conclusions. In my opinion, it should be number 4. Conclusions.
*****Sorry for the mistake. The number 5 has been replaced by 4 in conclusions. Please see the line 633.
I recommend that the authors consider revising the Conclusions in light of possible changes in the wording of the study objectives and the significant volume of results presented.
It is recommended that this section begin with a single sentence that indicates to the reader whether the null hypothesis of the study is confirmed or rejected. In the next two - three sentences should be formulated in the passive tense according to the important conclusions for fundamental science and for practice. E.g.: It was found that...
*****Thank you so much for your insightful suggestions. Please see the lines 635-640.
Based on the above, I propose minor revision.

Round 2
Reviewer 1 Report
Comments and Suggestions for Authors
Because the authors have revised this manuscript well, reflecting the opinions of reviewers, it is believed that it will be possible to publish it in this journal.
Author Response
Manuscript Number: foods-2956886
Research title: Effects of different phenolic compounds on the redox state of myoglobin and prevention of discoloration, lipid and protein oxidation of refrigerated longtail tuna (Thunnus tonggol) slices.
Thank you so much for the comments.